# Physiological stress and higher reproductive success in bumblebees are both associated with intensive agriculture

Tatjana Krama[1], Ronalds Krams[1,2], Maris Munkevics[3],
Jonathan Willow[2], Sergejs Popovs[1], Didzis Elferts[3], Linda Dobkeviča[3],
Patrīcija Raibarte[3], Markus Rantala[4], Jorge Contreras-Garduño[5] and
Indrikis A. Krams[1,3,6]

[1] Daugavpils University, Daugavpils, Latvia
[2] Estonian University of Life Sciences, Tartu, Estonia
[3] University of Latvia, Riga, Latvia
[4] University of Turku, Turku, Finland
[5] Universidad Nacional Autónoma de México, Morelia, Mexico
[6] University of Tartu, Tartu, Estonia

Corresponding author
Indrikis A. Krams,
indrikis.krams@ut.ee

## ABSTRACT

Free-living organisms face multiple stressors in their habitats, and habitat quality often affects development and life history traits. Increasing pressures of agricultural intensification have been shown to influence diversity and abundance of insect pollinators, and it may affect their elemental composition as well. We compared reproductive success, body concentration of carbon (C) and nitrogen (N), and C/N ratio, each considered as indicators of stress, in the buff-tailed bumblebee (*Bombus terrestris*). Bumblebee hives were placed in oilseed rape fields and semi-natural old apple orchards. Flowering season in oilseed rape fields was longer than that in apple orchards. Reproductive output was significantly higher in oilseed rape fields than in apple orchards, while the C/N ratio of queens and workers, an indicator of physiological stress, was lower in apple orchards, where bumblebees had significantly higher body N concentration. We concluded that a more productive habitat, oilseed rape fields, offers bumblebees more opportunities to increase their fitness than a more natural habitat, old apple orchards, which was achieved at the expense of physiological stress, evidenced as a significantly higher C/N ratio observed in bumblebees inhabiting oilseed rape fields.

## INTRODUCTION

While pollination is one of the most important ecosystem services for ensuring crop productivity, both agricultural intensification and agricultural land expansion have negatively affected pollinator species and community diversity (*Kremen, Williams & Thorp, 2002*; *Garibaldi et al., 2013*; *Lever et al., 2014*). This is an important economic argument for conservation of local pollinators and their habitats (*Goulson, 2012*). However, despite the numerous conservation measures in place, insect pollinators remain

under constant stress, especially in areas of high-intensity agriculture, as a result of continuous changes in the landscape and microclimate, invasive pests and diseases, decreases in the amount of floral resources, and the detrimental effects of pesticides (*Goulson et al., 2015*; *Crall et al., 2018*). Moreover, recent studies have demonstrated that minor variations in vegetation traits significantly affect habitat quality, pollinator immunity, stress resistance and abundance (*Riddell & Mallon, 2006*; *Bartual et al., 2019*; *Jachuła, Denisow & Wrzesień, 2021*).

It is known that body elemental composition responds to environmental stress (*Sterner & Elser, 2002*; *Hawlena & Schmitz, 2010a*; *Huang, Wang & Ren, 2019*). Body carbon (C) and body nitrogen (N) concentrations follow to changes in environmental quality (*De Senerpont Domis et al., 2014*; *Janssens, Op de Beeck & Stoks, 2017*), environmental stability (*Janssens, Van Dievel & Stoks, 2015a*; *Schmitz, Rosenblatt & Smylie, 2016*; *Zhang et al., 2016*), individual development-associated stress (*Trakimas et al., 2019*; *Krams et al., 2020*), and the risk of predation (*Janssens, Van Dievel & Stoks, 2015b*; *Van Dievel, Janssens & Stoks, 2016*; *Krams et al., 2016*; *Zhang et al., 2016*; *Krams et al., 2021a*). Stress is supposed to increase glucocorticosteroid release, which heightens the intensity of metabolism (*Slos & Stoks, 2008*; *Hawlena & Schmitz, 2010a*, *2010b*; *Krams et al., 2013a*, *2013b*). This typically leads to higher consumption of carbohydrates (*Hawlena & Schmitz, 2010a*; *Rinehart & Hawlena, 2020*), the increased concentration of body C, reduced concentration of body N, and a higher C/N ratio (*Sterner & Elser, 2002*; *Hawlena & Schmitz, 2010a*, *2010b*; *Trakimas et al., 2019*; *Van Dievel, Janssens & Stoks, 2020*).

In pollinator research, ecological stoichiometry has been used to study relationships between the chemical content and quality of bee food, and the consequences these may cause on the individual development of bees and the diversity of local pollinator communities (*Filipiak & Weiner, 2017*; *Filipiak et al., 2017*). The maintenance of stoichiometric balance is important because inconsistency between the chemical composition of an organism's tissues and that of its food sources strongly affects the organism's life history traits, including fitness (*Filipiak, 2018*; *Filipiak, Woyciechowski & Czarnoleski, 2021*). This suggests that the availability of suitable food and its elemental composition may influence the stoichiometric balance of individual pollinators (*Simanonok & Burkle, 2020*). The stoichiometric phenotype of pollinators has been shown to affect plant–herbivore interactions, the diversity of local communities, and the quality of pollination services performed (*Filipiak & Weiner, 2017*; *Kämper et al., 2017*).

The concentration of body C and N in bumblebees has been studied to understand the variation of C and N among social castes and sexes of the buff-tailed bumblebee (*Bombus terrestris*) in an agricultural landscape (*Krams et al., 2021b*). To understand the suitability of ecological stoichiometry based on variations of body C and N and the C/N ratio as indicators of bumblebee stress, comparative research should be done to compare the body C and N of bumblebees in habitats resembling natural ones and areas of high-intensity agriculture. A recent study shows that close proximity to forests favors bumblebee abundance and diversity because forests support wild pollinators also in nearby agricultural landscapes (*Sõber et al., 2020*). Due to the presence of several layers of

vegetation in apple orchards, we considered this to be a more natural habitat than oilseed rape fields.

In this study, we investigated whether habitat affects ecological stoichiometry in the bodies of *B. terrestris* adults. We predicted that *B. terrestris* individuals living in more natural habitats such as old apple orchards grown with no pesticides used should have higher body N concentrations, lower body C concentrations, and lower C/N ratios than *B. terrestris* individuals inhabiting intensive agricultural landscapes such as oilseed rape fields. Apple orchards consist of trees, and habitats containing more trees can be considered as habitats supporting higher biodiversity and abundance of wild bumblebees (*Martínez-Sastre, Miñarro & García, 2020*; *Sõber et al., 2020*). A previous study (*Krams et al., 2021b*) showed that body N concentration and body C/N ratio depend on *B. terrestris* caste, sex and queen age, while body C concentration does not significantly differ between bumblebee types (castes, ages and between young and ovipositing queens). Based on these results (*Krams et al., 2021b*), we predicted that workers and young queens might have higher N concentrations and lower C/N ratios than ovipositing queens and males. However, the flowering period of oilseed rape is estimated to reach 1 month (*d'Andrimont et al., 2020*) or longer in northern conditions as are present in Latvia, whereas apple trees bloom for only up to 10 days (*Tromp, 1976*). This suggests that bumblebee colonies can have access to larger amounts of food resources in oilseed rape fields than in apple orchards.

## MATERIALS AND METHODS

### Study sites and insects

The study was carried out using commercially grown *B. terrestris* (Biobest Group NV, Westerlo, Belgium). At the beginning of May 2020, we placed *B. terrestris* hives in two contrasting habitats in southeast Latvia near Krāslava: oilseed rape fields and apple orchards. We had a total of 50 hives across 28 oilseed rape fields and 22 hives across 11 apple orchards. At each study site, we placed one to three colonies, each protected against ant and rodent attacks by attaching sticky tape around the lower parts of each hive. We weighed each hive and removed the plastic container with a feeding solution attached to each hive by the producer. We regularly observed each colony to make sure the colonies were in good condition.

The apple orchards were located on private properties, and their size ranged from 2 to 4 ha. The orchards were partially surrounded by semi-natural forest vegetation in the form of mixed-species unmanaged patches of alders, oaks, pines, spruces, and hedgerows. The apple orchards represented old, semi-natural ecosystems with low commercial value. The average age of apple trees was 44.24 ± 6.12 years. The ground level was covered by blackcurrant (*Ribes nigrum*), redcurrant (*Ribes rubrum*) and gooseberry (*Ribes divaricatum*) bushes and grasses typical for local meadows, and cut for hay twice during spring–summer.

The nearest oilseed rape fields were located 3–6 km away from the apple orchards. The fields were c. 20–50 ha in size and were located in a mosaic of landscape as described
earlier (*Krams et al., 2021b*). We chose our study sites so that bumblebee hives were located between fields of spring- and winter oilseed rape.

We collected hives for their content analyses at the beginning of July just as they reached generative offspring production phase. The collection was done similarly as described in our earlier work (*Krams et al., 2021b*). Prior to collection we closed exit holes of the hive for 7–9 h and hives were stored in a freezer at −84 °C (Angelantoni Lifescience, Massa Martana, Italy).

## Nest variables

For each nest, we checked for the presence of ovipositing queens, and recorded the total number of hatched and unhatched cells. Since all hives contained males and young queens, all pupae were assumed to be next-generation offspring, and were thus included as a measure of reproductive success. The number of worker- and queen cells reflect reproductive effort, and since reproduction is costly, these numbers can be used to estimate the levels of reproduction-associated stress.

## Bumblebee body mass, body C and N concentration

We dried the bumblebees at 65 °C for 72 h, and weighed individual insects using a Precisa semi-micro balance (ES 225SM-DR; Precisa Gravimetrics AG, Dietikon, Switzerland) to obtain their dry mass. Pollen was removed from the body or workers before we dried and weighed bumblebees. Individual bumblebees were homogenized and we obtained their whole body C and N mass percentage using the element analyzer EuroVector EA3000 (Eurovector Srl, Pavia, Italy) (*Krams et al., 2020*, *2021a*). Samples of C and N concentrations were measured for each bumblebee. We had 50 ovipositing queen samples (22 collected from apple orchards, 28 from oilseed rape fields), 52 young queen samples (27 from apple orchards, 25 from oilseed rape fields), 63 worker samples (28 from apple orchards, 35 from oilseed rape fields), and 55 male samples (29 from apple orchards, 26 from oilseed rape fields).

## Statistics

We used the Gamma (with log link) generalized linear mixed-effects models (GLMMs) using habitat, bumblebee sex and interaction between habitat and sex as independent variables, and body mass, body C, and C/N ratio as dependent variables as those variables showed problems with heterogeneity. Linear mixed-effects models (LMEs) with the same independent variables were used for body N concentration. In all four models, Site identity (ID) and Hive ID were used as nested random factors.

A Poisson (with log link) GLMM was used to assess the effect of habitat (independent variable) on the number of intact worker cocoons, eclosed worker cocoons, total number of worker cocoons, intact queen cocoons, eclosed queen cocoons, total number of queen cocoons, number of pollen cells and number of wax cells (dependent variables). As the models showed overdispersion, Hive ID was added as the observation-level random effects, in addition to Site ID that was also set as a random factor in all eight Poisson GLMMs. We performed post-hoc comparison between bumblebee types (workers, young queens,

ovipositing queens, males) using pairwise Tukey-adjusted comparison of estimated marginal means from the model. Overall significance of each factor and their interaction was expressed as analysis of deviance tables (Type II Wald chi-square tests) for Gamma and Poisson GLMM, and analysis of variance table with Satterthwaite's method for LME. For the Gamma and Poisson GLMM, post-hoc test comparisons were evaluated using $z$-value, but $t$-value for LME. All analyses were performed in R v4.0.4 (*R Core Team, 2021*) libraries lme4 (*Bates et al., 2015*) for GLMM and LME models, lmerTest (*Kuznetsova, Brockhoff & Christensen, 2017*) for obtaining $p$-values of models, and emmeans (*Lenth, Buerkner & Herve, 2021*) for post-hoc tests. The significance threshold for all tests was set at $p = 0.05$.

# RESULTS

## Body mass

The body mass was significantly affected by bumblebee types ($\chi^2_{(3)} = 693.27$, $p < 0.0001$; Fig. 1). However, body mass was not affected by habitat ($\chi^2_{(1)} = 1.42$, $p = 0.23$) nor habitat–bumblebee type interaction ($\chi^2_{(3)} = 4.35$, $p = 0.23$).

Regarding specimens collected from apple orchards, ovipositing queens were significantly heavier than workers (mass ratio = 4.89 (SE = 0.499), $z = 15.549$, $p < 0.0001$), males (mass ratio = 2.45 (SE = 0.246), $z = 8.952$, $p < 0.0001$), and young queens (mass ratio = 1.44 (SE = 0.147), $z = 3.544$, $p = 0.0203$). Young queens were significantly heavier than workers (mass ratio = 3.40 (SE = 0.297), $z = 14.015$, $p < 0.0001$), and males (mass ratio = 1.71 (SE = 0.144), $z = 6.336$, $p < 0.0001$). Males were significantly heavier than workers (mass ratio = 1.99 (SE = 0.164), $z = 8.384$, $p < 0.0001$).

Regarding specimens collected from oilseed rape fields, ovipositing queens were also significantly heavier than workers (mass ratio = 4.34 (SE = 0.357), $z = 17.829$, $p < 0.0001$), males (mass ratio = 2.75 (SE = 0.240), $z = 11.592$, $p < 0.0001$), and young queens (mass ratio = 1.49 (SE = 0.134), $z = 4.494$, $p = 0.0004$). Young queens were significantly heavier than workers (mass ratio = 2.90 (SE = 0.237), $z = 13.053$, $p < 0.0001$), and males (mass ratio = 1.84 (SE = 0.166), $z = 6.766$, $p < 0.0001$). Males were also significantly heavier than workers (mass ratio = 1.58 (SE = 0.126), $z = 5.702$, $p < 0.0001$).

## Body carbon

We observed significant differences in body C concentration between bumblebee types ($\chi^2_{(3)} = 94.15$, $p < 0.0001$; Fig. 2) and habitats ($\chi^2_{(1)} = 41.75$, $p < 0.0001$; Fig. 2), while interaction of habitat and bumblebee type was not statistically significant ($\chi^2_{(3)} = 7.44$, $p = 0.0651$).

Regarding specimens collected from apple orchards, ovipositing queens had higher body C concentration than males (concentration ratio = 1.04 (SE = 0.008), $z = 4.702$, $p = 0.0002$), and workers (concentration ratio = 1.05 (SE = 0.010), $z = 4.774$, $p < 0.0001$). Young queens also had higher body C concentration than workers (concentration ratio = 1.04 (SE = 0.009), $z = 4.955$, $p < 0.0005$), and males (concentration ratio = 1.03 (SE = 0.007), $z = 4.741$, $p < 0.0001$). Meanwhile body C concentration in specimens from

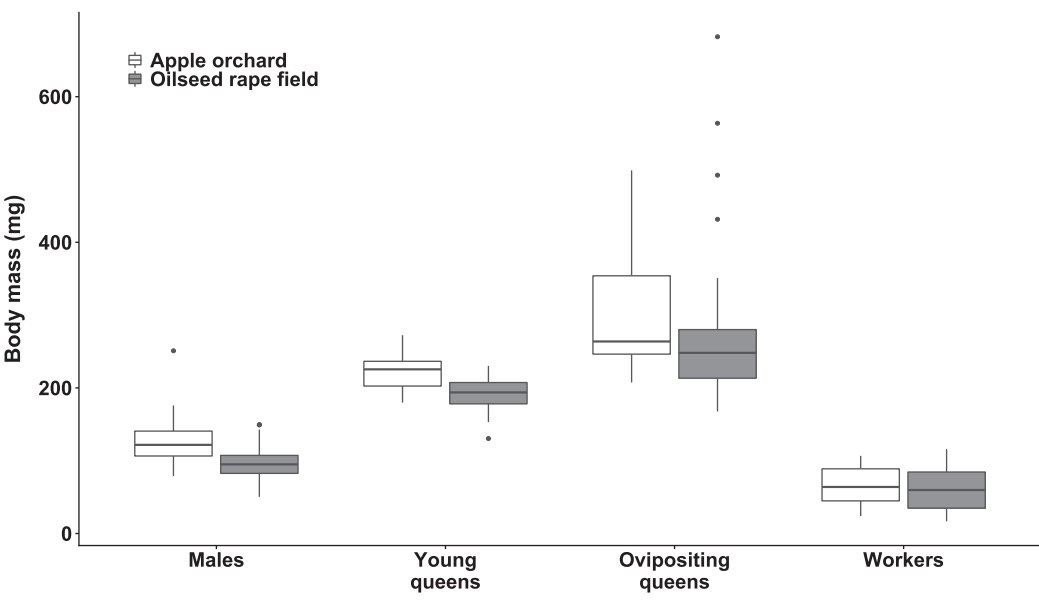

**Figure 1 Dry body mass of bumblebee males, young queens, ovipositing queens and workers.** Box plots showing dry body mass of males, young queens, ovipositing queens and workers of *Bombus terrestris* collected from apple orchards and oilseed rape fields.

apple orchards did not differ significantly between males and workers, and between young and ovipositing queens.

Regarding specimens collected from oilseed rape fields, ovipositing queens had higher body C concentration than males (concentration ratio = 1.04 (SE = 0.008), $z = 4.883$, $p < 0.0001$), and workers (concentration ratio = 1.07 (SE = 0.001), $z = 7.795$, $p < 0.0001$). Young queens had higher body C concentration than workers (concentration ratio = 1.05 (SE = 0.009), $z = 5.659$, $p < 0.0001$). In addition, males had higher body C concentration that workers (concentration ratio = 1.03 (SE = 0.008), $z = 3.676$, $p = 0.0127$), but body C concentration was not significantly different between queen age groups, and between young queens and males.

Specimens collected from apple orchards had significantly higher body C concentration, within all bumblebee types compared to specimens collected from oilseed rape fields (all $p$s < 0.0001).

### Body nitrogen

We observed significant differences in body N concentration between bumblebee types ($F_{(3, 199.48)} = 31.12$, $p < 0.0001$; Fig. 3) and habitats ($F_{(1, 6.92)} = 36.70$, $p < 0.0006$; Fig. 3). The interaction between habitat and bumblebee types was also significant ($F_{(3, 199.96)} = 9.96$, $p < 0.0001$).

Regarding specimens collected from orchards, workers had higher body N concentration than ovipositing queens (estimated difference = 2.04 (SE = 0.35), $t_{(173)} = 5.741$, $p < 0.0001$), and males (estimated difference = 2.67 (SE = 0.28), $t_{(180)} = 9.49$,

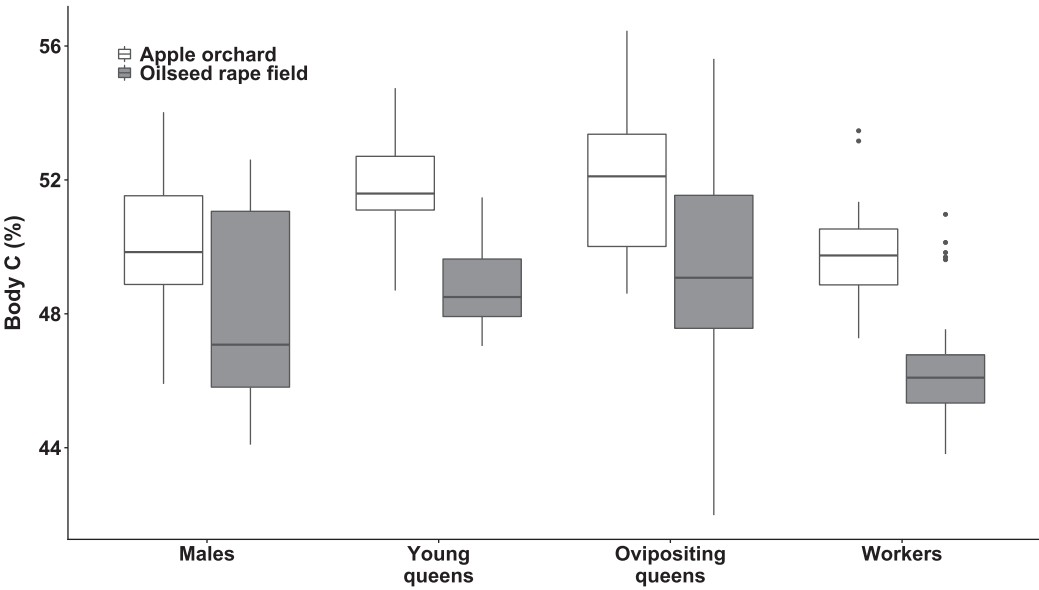

**Figure 2 Body carbon concentration of bumblebee males, young queens, ovipositing queens and workers.** Box plots showing body carbon concentrations of males, young queens, ovipositing queens and workers of *Bombus terrestris* collected from apple orchards and oilseed rape fields. C = carbon.

$p < 0.0001$). Young queens had higher body N concentration than males (estimated difference = 1.77 (SE = 0.06), $t_{(197)} = 5.863$, $p < 0.0001$).

Regarding specimens collected from oilseed rape fields, young queens had higher body N concentration compared to ovipositing queens (estimated difference = 1.47 (SE = 0.32), $t_{(209)} = 4.617$, $p < 0.0005$). In addition, workers had higher body N concentration than ovipositing queens (estimated difference = 1.22 (SE = 0.29), $t_{(207)} = 4.618$, $p < 0.003$).

Queens and workers from apple orchards had higher body N concentrations compared to those collected from oilseed rape fields (all $ps < 0.0007$), whereas males did not differ significantly in body N between apple orchards and oilseed rape fields.

## C/N ratio

We observed significant differences in the C/N ratio between bumblebee types ($\chi^2_{(3)} = 136.76$, $p < 0.0001$; Fig. 4) and habitats ($\chi^2_{(1)} = 14.79$, $p < 0.0002$; Fig. 4). The interaction between habitat and bumblebee types was also significant ($\chi^2_{(3)} = 19.33$, $p < 0.0003$).

Regarding specimens collected from apple orchards, body C/N ratio was significantly higher in males compared to workers (value ratio = 1.414 (SE = 0.054), $z = 9.022$, $p < 0.0001$) and young queens (value ratio = 1.247 (SE = 0.052), $z = 5.286$, $p < 0.0001$). Ovipositing queens also had higher body C/N ratio than workers (value ratio = 1.338 (SE = 0.069), $z = 5.633$, $p < 0.0001$), and young queens (value ratio = 1.180 (SE = 0.063), $z = 3.092$, $p = 0.0107$).

Regarding specimens collected from oilseed rape fields, ovipositing queens had higher body C/N ratios than workers (value ratio = 1.35 (SE = 0.05), $z = 7.43$, $p < 0.0001$), males

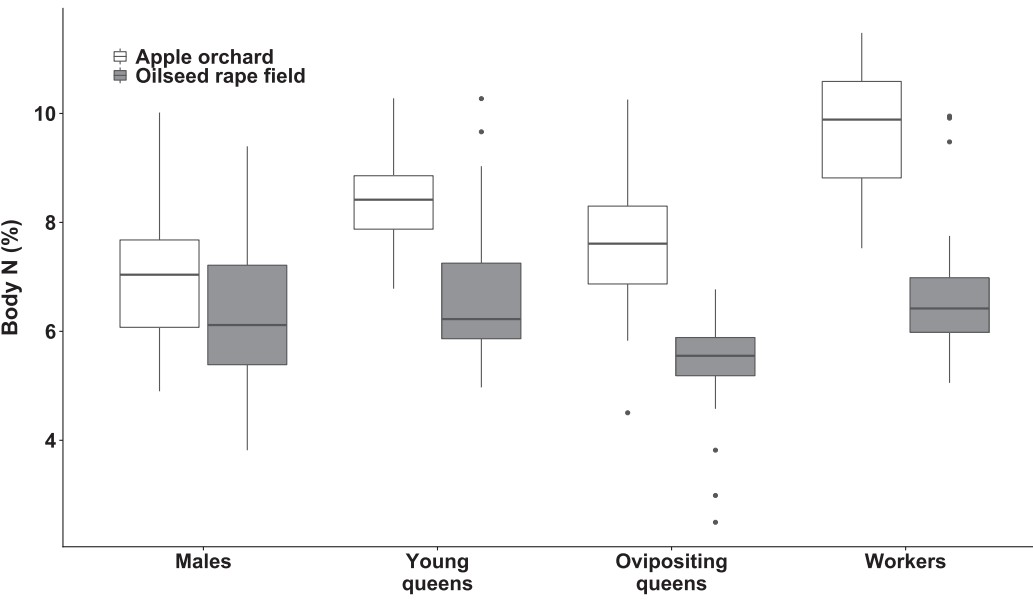

**Figure 3 Body nitrogen concentration of bumblebee males, young queens, ovipositing queens and workers.** Box plots showing body nitrogen concentrations of males, young queens, ovipositing queens and workers of *Bombus terrestris* collected from apple orchards and oilseed rape fields. N = nitrogen.

(value ratio = 1.19 (SE = 0.05), $z$ = 3.96, $p$ < 0.005), and young queens (value ratio = 1.33 (SE = 0.06), $z$ = 6.49, $p$ < 0.0001). Additionally, males had significantly higher body C/N ratio than worker bumblebees (value ratio = 1.14 (SE = 0.05), $z$ = 3.29, $p$ < 0.047).

Young queens, ovipositing queens, and workers from apple orchards had significantly lower body C/N ratios compared to corresponding caste members collected from oilseed rape fields (all $p$s < 0.0001), whereas males did not differ between habitats significantly.

## Number of worker- and queen cocoons

Regarding *B. terrestris* nests collected from oilseed rape fields, the number of worker cocoons eclosed ($\chi^2_{(1)}$ = 8.56, $p$ = 0.003, Fig. 5A) and total numbers of worker cocoons ($\chi^2_{(1)}$ = 3.89, $p$ = 0.049, Fig. 5B) were significantly higher than in nests collected from apple orchards. There was no significant difference between habitats, regarding number of queen cocoons eclosed ($\chi^2_{(1)}$ = 1.21, $p$ = 0.271, Fig. 5C), total number of queen cocoons ($\chi^2_{(1)}$ = 1.15, $p$ = 0.284, Fig. 5D) and the number of worker- ($\chi^2_{(1)}$ = 2.10, $p$ = 0.148) or queen cocoons intact ($\chi^2_{(1)}$ = 2.58, $p$ = 0.108).

## Number of pollen- and wax cells

We observed a significantly higher number of pollen cells ($\chi^2_{(1)}$ = 11.3, $p$ < 0.001, Fig. 5E) in *B. terrestris* nests collected from oilseed rape fields, compared to nests collected from apple orchards, but there was no significant difference for the number of wax cells ($\chi^2_{(1)}$ = 3.11, $p$ = 0.078, Fig. 5F).

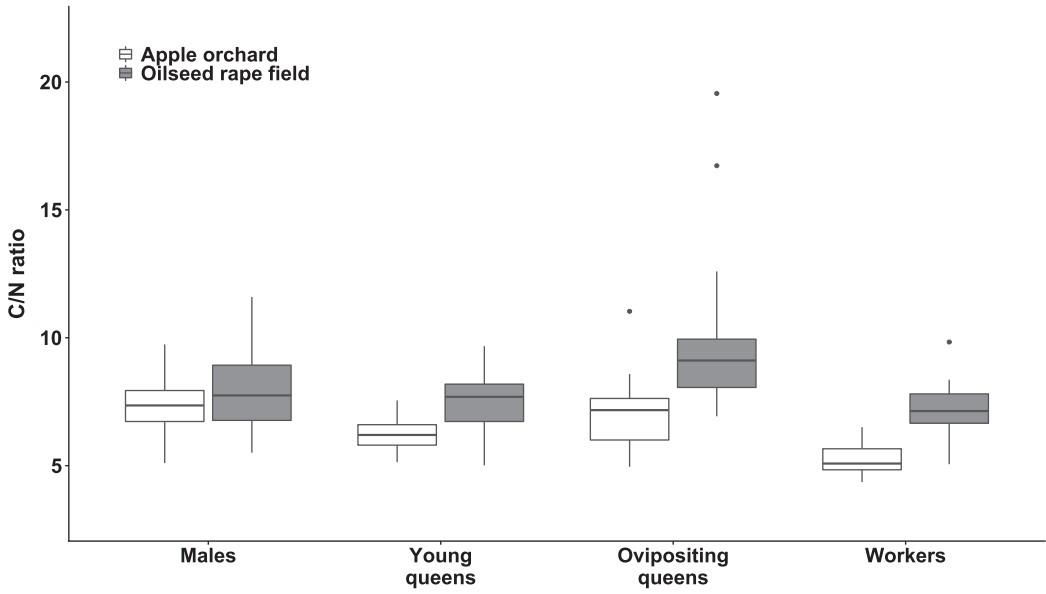

**Figure 4 Body carbon to nitrogen ratio of bumblebee males, young queens, ovipositing queens and workers.** Box plots showing body carbon:nitrogen ratio of males, young queens, ovipositing queens and workers of *Bombus terrestris* collected from apple orchards and oilseed rape fields. C = carbon, N = nitrogen.

# DISCUSSION

Theory predicts that environmental change and associated physiological reactions should alter the macronutrient demands of stressed organisms and affect their development and reproductive strategies (*Hawlena & Schmitz, 2010b*; *Schmitz, Rosenblatt & Smylie, 2016*; *Krams et al., 2020*). In this study, we showed that while *B. terrestris* body mass differed between castes, these differences were not present between specimens collected from apple orchards and oilseed rape fields. However, body C and N concentrations of bumblebee castes were generally lower in specimens collected from oilseed rape fields compared to apple orchards, while the observed C/N ratios were lower in specimens collected from apple orchards compared to oilseed rape fields. Specifically, specimens of young queens, ovipositing queens and workers collected from apple orchards showed lower C/N ratios and higher body N concentrations compared to corresponding specimens collected from oilseed rape fields. Since a higher C/N ratio is often considered to be an indicator of stress, we suggest that all castes of *B. terrestris* except males likely experienced functionally higher stress levels in oilseed rape fields than in apple orchards.

Apple trees and oilseed rape are both important agricultural plants. Oilseed rape cultivation severely depletes the soil of nutrients, and thus requires large amounts of fertilizers, which is not typical for cultivation of apple trees (*Colnenne, 1998*; *Dubousset, Etienne & Avice, 2010*). This suggests that oilseed rape invests significant resources into intense growth, as well as the flowering that provides pollinators with food resources. Therefore, oilseed rape is considered a nectar- and pollen rich crop (*Berrocoso et al., 2015*; *Bertazzini & Forlani, 2016*). Nectar is important to adult bumblebees as a source of carbohydrates for sustaining flight and foraging. Pollen is crucial to the development of

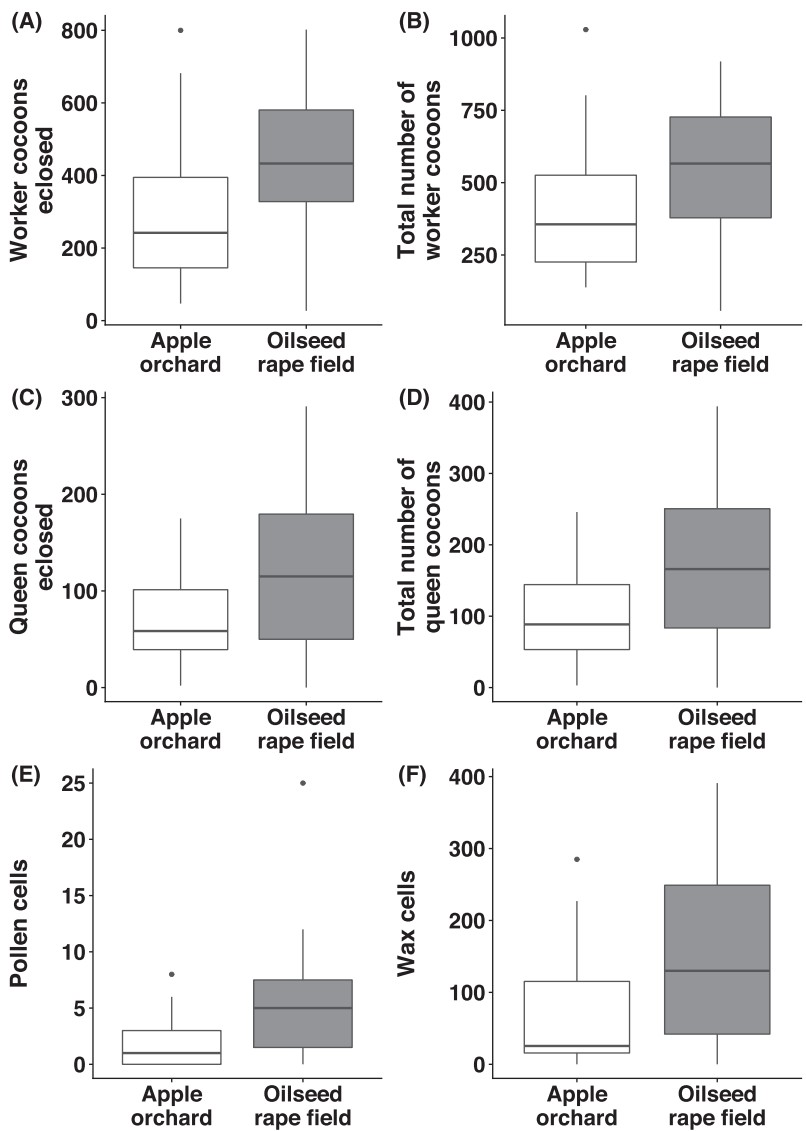

**Figure 5** **Reproductive parameters of bumblebees in old apple orchards and oilseed rape fields.** For *Bombus terrestris* nests collected from apple orchards and oilseed rape fields, box plots showing the number of worker cocoons eclosed (A), total number of worker cocoons (intact and eclosed; B), number of queen cocoons eclosed (C), total number of queen cocoons (intact and eclosed; D), number of pollen cells (E), and number of wax cells (F).               

larvae, as it is the principal source of proteins (*Kriesell, Hilpert & Leonhardt, 2017*). Both of these food resources are abundant over the course of 4 to 6 weeks in oilseed rape agroecosystems (depending on cultivar) (*Woodcock et al., 2016*). The enormous flower density and abundant nectar and pollen resources provided by oilseed rape benefit such oilseed rape-foraging bumblebee species as *B. terrestris* (*Hoyle, Hayter & Cresswell, 2007*; *Kleijn et al., 2015*). Mass-flowering of oilseed rape attracts pollinators, potentially resulting in underpollination of other local plant species such as native wildflowers (*Holzschuh et al., 2011*). In contrast, flowering apple trees are highly attractive to bumblebees for a maximum of 10 days during their mass-flowering period in our study region. Accordingly,

bumblebee colonies may grow faster when located in- or around oilseed rape fields; subsequently, the colonies will start producing the reproductive generation (males and young queens) earlier than would occur in apple orchards (*Woodcock et al., 2016*). Since life history traits reflect the differential allocation of different resources to competing life functions (*Speakman & Garratt, 2014*), trade-offs between higher resource abundance and more intense reproductive investments may result in higher stress levels caused by the tremendous reproductive effort of bumblebees in oilseed rape fields.

Our data show that higher C/N ratios in bumblebees collected from oilseed rape fields coincides with much higher reproductive output in this habitat compared to apple orchards. The higher total number of worker cocoons, and larger amounts of food resources (pollen cells), were observed in hives collected from oilseed rape fields, compared to those collected from apple orchards. More food evidently allowed bumblebees in oilseed rape fields to invest in reproduction, resulting in a significantly higher number of workers and young queens produced. Reproduction comes at the cost of affecting lifespan, immunity, and other life history traits (*Lawniczak et al., 2007*; *Harshman & Zera, 2007*), while extra physical activities may be detrimental for longevity (*Tanaka et al., 2019*; *Hayashi & Kezuka, 2020*). The production of sex hormones can also mediate the cost of reproduction by trading off higher investment in reproduction against decreased investment into somatic functions (*Harshman & Zera, 2007*).

Food quality and energetic value are of particular importance in life history decisions, and can greatly enhance an organism's investment in reproduction and fitness when available *ad libitum* (*Krams et al., 2015*, *2017*). Thus, food availability often improves reproductive success, which we may have observed here with regard to bumblebees inhabiting oilseed rape fields. However, as investments in reproduction often compete with the immune system for the same resources, high reproductive investment and fitness-gain can come at the expense of stress reflected in higher C/N ratios found in workers, ovipositing queens, and young queens inhabiting oilseed rape agroecosystems.

The absence of a significant difference between the C/N ratios of males in apple orchards and oilseed rape fields supports the idea that a higher C/N ratio may be due to an investment in reproduction, as well as the associated high-intensity foraging behavior observed in ovipositing queens, young queens and workers in oilseed rape fields. It is important to note that the availability of nectar and pollen in oilseed rape fields sharply ends before the end of bumblebee reproductive season (*Carvell et al., 2011*; *Williams, Regetz & Kremen, 2012*; *Woodcock et al., 2016*), which is not the case in apple orchards where bumblebees rely on many other plants besides apple trees. It has been found that while abundance of foraging resources is of high importance for worker production in bumblebees (*Westphal, Steffan-Dewenter & Tscharntke, 2009*; *Woodcock et al., 2016*), queen production and population sustainability depends on resource availability during the whole breeding season (*Williams, Regetz & Kremen, 2012*; *Woodcock et al., 2016*). Our results fully support these findings showing higher investment in worker production in *B. terrestris* breeding in oilseed rape fields and similar effort in queen production in apple orchards and oilseed rape fields. Future research should measure lifespan and investigate the immune system of bumblebees reproducing in different habitats, including

the most optimal habitats for pollen and nectar availability to better understand the role of ecological factors such as floral phenology in trade-offs between life history traits.

Our data suggest that apple orchards may be an optimal habitat for certain bumblebees. This may be due to the higher proportion of trees having a positive impact on biodiversity and abundance of bumblebees (*Sõber et al., 2020*). However, the flowering season for apple trees is relatively short, subsequently increasing *B. terrestris*'s body N demand, as this element is found in muscles required for longer foraging flights. In contrast, bumblebees can find food within the vicinity of their hives in- and around oilseed rape fields, decreasing their body N demand (*Krams et al., 2021b*). This shows that flora phenology must be taken into account to better understand the role of habitat quality on survival and reproductive strategies of pollinators.

It is important to note that bumblebees and their hives may be relatively more exposed to direct sunlight in oilseed rape agroecosystems than in apple orchards. This could lower the need for excess carbohydrate-rich food, and decrease their body C concentration. Under lower ambient temperatures, or no access to direct sunlight, bumblebees must warm up to fly and forage (*Bujok et al., 2002*; *Seeley et al., 2003*). Bumblebee flight muscles must reach at least 30 °C in order for the bumblebee to become airborne (*Goller & Esch, 1990*). Generation of heat can be achieved *via* flight muscle shivering, as well as in the absence of shivering, by means of receiving action potentials (*Esch, Goller & Heinrich, 1991*). If bumblebee hives are more exposed to the direct sunlight, these muscle-associated heat generating activities may be less needed, explaining lower body N concentration in bumblebees collected from oilseed rape fields. More favorable ambient temperatures could also explain the lower body C concentrations we observed in bumblebees collected from oilseed rape fields, as less carbohydrate-based fuel is required for thermogenesis in warmer habitats.

Finally, lower body N concentrations and higher C/N ratios in specimens collected from oilseed rape fields, compared to apple orchards, may be explained by greater investments in reproduction in terms of offspring number, and less so in their somatic growth, because bumblebees could potentially reach larger sizes when living in oilseed rape agroecosystems. It was observed that grasshoppers consume and process more proteins under higher ambient temperatures (*Schmitz, Rosenblatt & Smylie, 2016*), and we suggest that in bumblebees a similar relationship may result in more rapid development of workers and higher reproductive output, as we observed in hives collected from oilseed rape fields. Therefore, relationships between investment in offspring production, temperature effects, and body C- and N concentrations should be investigated in future research.

## CONCLUSIONS

We found significant differences in reproductive success, body C- and N concentrations, and C/N ratio of *B. terrestris* types (workers, young queens, ovipositing queens, males) in two agricultural habitats. Surprisingly, our results show that reproductive output was significantly higher in oilseed rape fields than in apple orchards, the latter being considered a more natural habitat due to the presence of several layers of vegetation. On the other hand, physiological stress, indicated by the higher C/N ratios of queens and workers, was

lower in apple orchards. This suggests that bumblebees can achieve higher fitness when reproducing in oilseed rape fields. Our study highlights a positive impact of high-intensity oilseed rape cultivation, particularly the achievement of meeting the nutritional requirements of animals (*Knutie, Chaves & Gotanda, 2019*), in this case the bumblebee *B. terrestris*. We suggest that approaches based on ecological stoichiometry may be instrumental in building on the knowledge of how habitat quality can affect pollinators. In this study, we did not carry out any experimental tests to provide additional resources to manipulate the duration of food availability in the apple orchards. Although these tests could help validate our results, extra sucrose solution cannot replace the whole quality of the habitat, consisting of different floral resources, microclimate and microorganisms, and pollen. While we plan to perform such experimental trials in the future, they require extra caution.

## ACKNOWLEDGEMENTS

The authors thank Professor Māris Kļaviņš for supporting this project at the stage of lab analyses.

### Funding

This work was supported by the Latvian Council of Science (grants lzp-2018/1-0393, lzp-2018/2-00057, lzp-2020/2-0271, lzp-2021/1-0277), and the European Union, European Regional Development Fund (Estonian University of Life Sciences ASTRA project "Value-chain based bioeconomy" project number 2014-2020.4.01.16-0036). The funders had no role in study design, data collection and analysis, decision to publish, or preparation of the manuscript.

### Grant Disclosures

The following grant information was disclosed by the authors:
Latvian Council of Science: lzp-2018/1-0393, lzp-2018/2-00057, lzp-2020/2-0271 and lzp-2021/1-0277.
Daugavpils University: 14-95/2022/17.
European Union, European Regional Development Fund: 2014-2020.4.01.16-0036.

### Competing Interests

The authors declare that they have no competing interests.

### Author Contributions

- Tatjana Krama conceived and designed the experiments, performed the experiments, analyzed the data, authored or reviewed drafts of the paper, and approved the final draft.
- Ronalds Krams conceived and designed the experiments, performed the experiments, analyzed the data, prepared figures and/or tables, authored or reviewed drafts of the paper, and approved the final draft.

- Maris Munkevics performed the experiments, analyzed the data, prepared figures and/or tables, authored or reviewed drafts of the paper, and approved the final draft.
- Jonathan Willow performed the experiments, authored or reviewed drafts of the paper, and approved the final draft.
- Sergejs Popovs performed the experiments, analyzed the data, authored or reviewed drafts of the paper, and approved the final draft.
- Didzis Elferts analyzed the data, prepared figures and/or tables, authored or reviewed drafts of the paper, and approved the final draft.
- Linda Dobkeviča analyzed the data, prepared figures and/or tables, authored or reviewed drafts of the paper, and approved the final draft.
- Patrīcija Raibarte conceived and designed the experiments, authored or reviewed drafts of the paper, and approved the final draft.
- Markus Rantala conceived and designed the experiments, prepared figures and/or tables, authored or reviewed drafts of the paper, and approved the final draft.
- Jorge Contreras-Garduño conceived and designed the experiments, prepared figures and/or tables, authored or reviewed drafts of the paper, and approved the final draft.
- Indrikis A. Krams performed the experiments, analyzed the data, authored or reviewed drafts of the paper, and approved the final draft.

## Data Availability

The data is available at Zenodo: Indrikis Krams. (2021). Ecological stoichiometry of bumblebees in agricultural landscapes [Data set]. Zenodo. https://doi.org/10.5281/zenodo.5128075.

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
