# Peer review of "Physiological stress and higher reproductive success in bumblebees are both associated with intensive agriculture"

_PeerJ, doi:10.7717/peerj.12953_

## Round 0.1 · original submission · Major Revisions

Both reviewers generally enjoyed your manuscript and found much to be excited about. However, both also pointed out several areas that need improvement. For example, there is evidence of copy/paste from other manuscripts (please address this), all sections need clarification and some statistical approaches need explanation, clarification, and in some cases need to be reconsidered.

Reviewer 1 ·

Excellent Review

This review has been rated excellent by staff (in the top 15% of reviews)
EDITOR COMMENT
This manuscript will be greatly improved by this reviewer's detailed, constructive and insightful comments, particularly by their advice for remedying some problems with the statistical approaches used to analyze the data.

Basic reporting

It’s always nice to see an ecological stoichiometry project in a terrestrial system, and this is one of a small handful of such papers focused on bee species that I am aware of.

There is a high degree of copy, pasting, and extremely minor editing/rearranging from previous articles from the group. In the methods this introduces errors to the manuscript (e.g., “flies”, L161, https://doi.org/10.1093/cz/zoaa070). In the introduction it results in the first and second half of the introduction having extremely different voices and depth of scientific background. In one paragraph describing the study design only the year and the temperature of freezing are changed, even though they are supposed to be two different studies performed in two different years. While I could understand for repeating methods across studies, the authors do not even refer to the other manuscripts in such a way as would be typical for similar methods or study designs (e.g., “methods were performed following Citation (Year), and thus described briefly here”). This is particularly stark in lines 62-78, 143-150, and part of the “Bumblebee Body C and N Content” heading. The degree to which it is done in this manuscript is simply unacceptable, and borders on plagiarism especially when considering that most of the copy and paste is from an article which was published this very week (https://doi.org/10.3389/fphys.2021.696689). Given that I found at least two sources here makes me concerned that there may be more I missed as well.

Experimental design

From reading the statistical methods, it seems that you used simple individual variable comparisons for each model as opposed to full models including all explanatory variables? This choice needs to be very carefully explained, as the models then do not reflect the study design and may be dramatically and incorrectly influencing the results. Further, on Gamma and Poisson distributions: Poisson makes sense to me for the latter models as those variables are clearly count data. I would add if you checked models for overdispersion and how you addressed that if present. Why was Gamma selected for the first set of models? Furthermore, there are extreme inconsistencies between the description of your analyses in the methods and the reported statistics. See line comments.

For the models that feed into your first paragraph in the Body Mass result (and Figure 1) it is unclear how you combined/split castes. It seems as though you lumped workers, gynes, and queens all together in some analyses but not others? Interchangeably using sex and caste for bumble bees is inappropriate; caste should be the focus given how biologically different workers vs. queens can be. Using sex for some analyses and caste for others makes understanding the results rather confusing. I’d suggest having the castes together as a factor with four levels (workers, gynes, queens, males). This would allow you to streamline your models, stats reporting, and effectively combine figures 1-3. Something like a Tukey HSD would then very easily determine differences among castes and between habitats.

Validity of the findings

The discussion and conclusion sections are insightful and well-written, though I anticipate significant changes will follow methodological clarity. I do think that the findings from this study could be interesting, and the discussion’s focus on the trade-offs between bee health and reproductive output is a high-point and could be an important finding in the bee world. Other authors have discussed these ideas, but the dataset presented here is uniquely poised to directly answer questions of stoichiometric trade-offs in a social insect. However, given the methodological issues as well as the presence of blatant copy/paste, it is not acceptable for publication.

Additional comments

Line Comments:
L50: Opening sentence focusing on freshwater resources and fisheries does not fit the narrative of the study well. The rest of the opening paragraph is sound. Alternatively, more comparison against previous non-pollinator ES work later in the discussion could help make this a more broadly-accessible and widely-read manuscript.
L73-78: Thank you for making the ES -> nutrition/health link. Most papers in this vein do not explicitly make this critical connection.
L80-89: Filipiak has a few other recent papers that would fit well here:
https://doi.org/10.3390/insects9030085 and https://doi.org/10.1038/s41598-020-79647-7
L80-100: While not framed specifically within ES, there’s a few other recent publications with %N in bee species which are likely to be relevant for you, and would help fill out this later section of the introduction: https://doi.org/10.1007/s00442-019-04577-9 and https://doi.org/10.1007/s13592-016-0480-4

L103-108: Your argument that the apple orchards would provide better resources than the oilseed rape fields would vary widely based on location and the type of apple orchard. For example, in other countries apple orchards are often highly intensive agricultural environments, much like how almond orchards can be extremely stressful for honey bee colonies. Thus, I think your hypotheses are dependent on the context of the study areas described in your methods, and I would consider moving your hypotheses into your methods section or providing some of that system-specific context alongside your hypotheses in the introduction. Furthermore, as an organism’s ecological stoichiometry is derived from their food source, it would help to discuss the stoichiometry and/or nutrition of the nectar and pollen sources present within and near orchards vs. rapeseed farms (you do mention this in the discussion; and it may be worth moving that here). While yes, complex habitats support a more diverse assemblage of bees, that relationship does not necessarily relate to ES.

L131: repeated ‘partially’

L143-150: This paragraph is copied nearly verbatim from another publication in your group, with only the year and temperature changing. I’d go through and ensure that these details are accurate: https://doi.org/10.3389/fphys.2021.696689

L161: Whole ‘flies’?

L172: bumblebee sex or caste? Later sentence says caste.

L180: “enclosed”

L182: Why was hive ID included for previous models but not these? I think it would be essential to include in these models, as reproductive output could depend strongly on the colony.

L195-299: The methods suggest that all models reported in this section used the same general framework and distributions. However, the test statistics reported vary widely suggesting a range of different analyses; I see chi-square, z statistics, F statistics, t statistics, as well as the omission of any such reporting (some with only a p value). Please review your models and revise.

L303-314: Likewise, I am not aware of a Poisson GLMM which returns a chi square statistic as the test statistic for individual parameters. Are these comparisons of different models? Please make clear in the methods.

L379: I don’t think the results clearly make this determination. As you discuss in the previous paragraph, bees were likely more stressed in rapeseed fields yet they had greater reproductive output. The arguments on trade-offs are much more closely aligned with your results.

L381-385: The phenology of the main flora of your habitats is an extremely important point, though I would expect the surrounding habitat would be more influential in the orchards due to the smaller patch size.

L389: Relating results to tropical flora does not seem particularly relevant to the system. The hypothesis presented in this paragraph is interesting though.

Minor thing:
“Submitted” articles are cited a few times. Make sure this lines up with PeerJ guidelines:
“Cite articles accepted for publication as 'in press'. Include in reference section and upload as a Supplemental file.
Cite work unpublished, in preparation or under review as 'unpublished data'. Supply the author's first initial and surname, and the year of the data collection, in the text citation and do not include the citation in the reference section. Example: (A Castillo, 2000, unpublished data).”

·

Excellent Review

This review has been rated excellent by staff (in the top 15% of reviews)
EDITOR COMMENT
This reviewer went above and beyond in recommending ways to clarify the study's methodology. Their recommendations will undoubtedly improve the manuscript.

Basic reporting

This is an interesting study that makes a novel contribution toward understanding how environmental stressors might affect body elemental ratios of carbon (C) and nitrogen (N). The study is nicely motivated by appropriate conceptual theory. The data appear to be appropriately analyzed. The MS is clearly written.

Experimental design

I have several comments and questions that need to be addressed in order to better judge the reliability of the data and the conclusions drawn from the study.

1) The MS claims that oilseed rape fields are more stressful environments than are apple orchards. But given that the study uses commercially reared (as opposed to wild bees) bees both kinds of habitat could be considered stressful to individuals that come from commercial sources. So, is there independent information or evidence from this or previous studies showing that commercial bees placed in these different kinds of environment are indeed differently stressed? For example, do this or other studies show elevated metabolism in one habitat relative to another, or altered immune-response in one habitat vs another? Offering more insight about the potential level of stress that these bees experience in the different environments would strengthen the case that body C:N differences are indeed due to stress.

2) Do C and N contents of pollen and nectar differ between oilseed rape and apples? Could this explain the differences in bee C:N between the habitats instead of stress experienced in the different fields? The fact that bee colonies in oilseed rape fields have higher fitness indicators might suggested that the stress hypothesis is not tenable.

3) How long were the colonies left out in the respective fields? The reason I ask is that if the colonies were out longer than the 10-day flowering period of apple orchards, where would the apple orchard bees get food for the remaining time in the field. If they get food elsewhere or from other plants, how much would this influence the measure of C:N for bee colonies placed in apple orchards and fitness measures?

4) Line 154: It is unclear what “post-freezing” means. Were the collected colonies placed in a freezer to kill all of the adult and developing bees before analysis?

5) The Results section lists a bunch of statistical analyses on various attributes of bees and bee colonies. But the Methods section provides extremely vague explanations of how the various attributes were measured.

How was bee body mass measured?

How were individuals processed in the lab to extract measures of C and N contents. Was pollen removed from the bodies of worker bees before analyses? Were gut contents removed from individuals before they were processed to measure body C:N?

What is the purpose of measuring the number of worker and queen cocoons and pollen and wax cells? What do more or less quantity of each of these various measures mean? Explaining this would help the general readership of PeerJ that is not familiar with bee colony life-history. Also explain better how these variables were measured and why? For example, are these fitness indicators?

Why were different caste members and sexes analyzed separately for C:N. Do each potentially face different stress? Please explain more the rationale for doing this.

The Discussion tends to be a bit more speculative about the potential that oilseed rape fields are more stressful than apple orchards, as opposed to the Introduction section which sets the contrast up as an a priori test of hypothesis about the consequences of different stress on C:N stoichiometry. Given my comments above, more information needs to be provided to offer a definitive a priori test of the stress hypothesis (i.e., rule out competing explanations). This would be ideal: I like tests of hypotheses. In the absence of being able to provide such information, the MS cannot provide a definitive test. Consequently it should be rewritten to be more speculative, that is, to explore the potential for the different habitats to be differentially stressful based on measure of C:N stoichiometry. This would be less ideal but perhaps the only option.

Validity of the findings

The findings could make a novel contribution to the field if my concerns of alternative explanations are addressed.

Additional comments

Minor comments:

First sentence of the Introduction is superfluous. Delete it as it has nothing to do with this study.

Line 102: “habitat quality” is vague. What defines habitat quality for bees?

Line 109: The MS refers to Krams et al. (submitted) here and in many places throughout the text. It would help the reader if the MS explained a bit more what that Krams study was about and what it found about C:N. that is germane to this study.

Line 405: This explanation is rather convoluted and hard to follow and contradictory to the main thesis of the MS. Especially so, because the C:N data presumably indicate that oilseed rape fields might be more stressful for bees, yet here the reasoning makes the opposite case that it may be a better environment than apple orchards for bees to be active. So which is it? A stressful or not so stressful environment?

---

## Round 0.2 · Minor Revisions

Both reviewers thank you for addressing their comments and concerns in your revised manuscript. However, Reviewer 1 remains concerned with your data analysis approach, particularly with how you grouped females.

Reviewer 1 ·

Basic reporting

Overall the manuscript is very close. I have a few remaining comments below.
The expanded detail and context in the introduction is also very helpful and improves the framing of the study.

Experimental design

Thank you for expanding the methodological detail, it is now much more clear which analyses were used.

Validity of the findings

I believe I must have been unclear in my response regarding the grouping of sex v. caste for analyses. My point is that as social insects, queens, gynes, and workers are going to have different body composition (as you demonstrate in Figures 2 and 3) and thus grouping the three female castes together as a single “female” level to compare against drones (Figure 1) misses a great deal of nuance. Similarly, having queens and gynes analyzed separately from workers misses important differences. For example, if you consider the values plotted for the C/N ratio panels (Figures 2D and 3D), “young queens” appears to be similar to workers whereas “oviposting queens” alone may be driving the reported difference between workers and queens. Likewise, drones may be different in their C/N composition from some castes of females but not others, as has been seen in other species of social insects (e.g., https://doi.org/10.1111/j.1365-2435.2009.01604.x , https://doi.org/10.1007/s00040-012-0220-3). Body mass is another variable where it does not make biological sense to group all females together, with the massive size variation from workers to queens that is possible even within single bumble bee colonies. As presented, the findings suggested from the analyses underlying Figures 1 and 2 may only exist as a result of the decision to group multiple castes of females together in different ways for different models. Performing your analyses with a caste factor with four levels (drone, worker, gyne, queen) would tease apart potential differences among them, simplify your analyses by reducing the number of models and figures, more accurately reflect their functional and biological differences, still answer your primary questions, and allow this manuscript to have greater reach among entomologists studying other social insects. Your response to a comment by the other reviewer shares the same rationale: “Moreover, workers, queens, males, young queens and ovipositing queens have different social, functional and reproductive roles in the hive and outside the hive. “

Additional comments

L110: Missing word: “Based these…”
L351-353: This sentence on the phenology of oilseed rape v. apple trees misses the habitat context: that the former is an agricultural monoculture and the latter is a more complex, patchy habitat with many other food resources and edge habitats.
L354-356: You did not measure rate of growth, only final reproductive output.
L356-359: Phrasing this as a question is strange as you answer the question earlier in the paragraph. Would be more effective and interesting to introduce the tradeoff idea here before the next paragraph: indication of stress in oilseed rape field, yet more reproductive output in oilseed rape fields.
L384-387, 389-396: I think the approach of trying to give a simple answer of which habitat is “better” doesn’t match the results. Individual bees are more/less stressed in one vs. the other, while colonies are more/less reproductive in one vs. the other. Both habitats are “better”, from a certain point of view. Clearly the truth is somewhere in the middle. Furthermore, a limited study where purchased colonies were only on the ground for a few months should be careful to make broad conclusions. The strength of this work is your ability to assess the tradeoffs of ES stress vs. reproductive output in bees, and I would focus on that in these first few paragraphs and then the closing of the discussion. It’s a truly novel contribution and I would really work to make that the focus of the discussion.
L398: Strike ‘unlimited’
L398-413: This hypothesis is interesting but, particularly after the revision, feels extremely disconnected from the rest of the manuscript. I’d either remove it or move a brief mention of the hypothesis (1 sentence?) to the following paragraph.

·

Basic reporting

Improved based on my comments in the original review.

Experimental design

Improved based on my comments in the original review.

Validity of the findings

Improved based on my comments in the original review.

---

## Round 0.3 · accepted · Accept

Thanks very much for carefully and successfully addressing the latest round of comments from Reviewer 1.